# Variations in the Distribution of Chl-*a* and Simulation Using a Multiple Regression Model

**DOI:** 10.3390/ijerph16224553

**Published:** 2019-11-18

**Authors:** Jiancai Deng, Fang Chen, Weiping Hu, Xin Lu, Bin Xu, David P. Hamilton

**Affiliations:** 1State Key Laboratory of Lake Science and Environment, Nanjing Institute of Geography and Limnology, Chinese Academy of Sciences, Nanjing 210008, China; wphu@niglas.ac.cn; 2Monitoring Center of Hydrology and Water Resources of Taihu Basin, Wuxi 214024, China; chenfang@tba.gov.cn (F.C.); xubin@tba.gov.cn (B.X.); 3Institute of Agricultural Resources and Environment, Jiangsu Academy of Agricultural Sciences, Nanjing 210014, China; lxdeng@126.com; 4Australian Rivers Institute, Griffith University, Queensland 4111, Australia

**Keywords:** Chl-*a* concentrations, water quality, spatiotemporal distribution, simulation, Lake Taihu, eutrophication

## Abstract

Chlorophyll *a* (Chl-*a*) is an important indicator of algal biomass in aquatic ecosystems. In this study, monthly monitoring data for Chl-*a* concentration were collected between 2005 and 2015 at four stations in Meiliang Bay, a eutrophic bay in Lake Taihu, China. The spatiotemporal distribution of Chl-*a* in the bay was investigated, and a statistical model to relate the Chl-*a* concentration to key driving variables was also developed. The monthly Chl-*a* concentration in Meiliang Bay changed from 2.6 to 330.0 μg/L, and the monthly mean Chl-*a* concentration over 11 years was found to be higher at sampling site 1, the northernmost site near Liangxihe River, than at the three other sampling sites. The annual mean Chl-*a* concentration fluctuated greatly over time and exhibited an upward trend at all sites except sampling site 3 in the middle of Meiliang Bay. The Chl-*a* concentration was positively correlated with total phosphorus (TP; *r* = 0.57, *p* < 0.01), dissolved organic matter (DOM; *r* = 0.73, *p* < 0.01), pH (*r* = 0.44, *p* < 0.01), and water temperature (WT; *r* = 0.37, *p* < 0.01), and negatively correlated with nitrate (NO_3_^−^-N; *r* = −0.28, *p* < 0.01), dissolved oxygen (DO; *r* = −0.12, *p* < 0.01), and Secchi depth (ln(SD); *r* = −0.11, *p* < 0.05). A multiple linear regression model integrating the interactive effects of TP, DOM, WT, and pH on Chl-*a* concentrations was established (*R* = 0.80, *F* = 230.7, *p* < 0.01) and was found to adequately simulate the spatiotemporal dynamics of the Chl-*a* concentrations in other regions of Lake Taihu. This model provides lake managers with an alternative for the control of eutrophication and the suppression of aggregations of phytoplankton biomass at the water surface.

## 1. Introduction

Phytoplankton plays an important role in regulating the energy available to higher trophic level consumers in aquatic ecosystems [1]. However, a rapid and substantial increase in phytoplankton biomass, especially cyanobacteria, often results in nuisance algal blooms, which threaten water quality and aquatic ecosystem health [2]. Chlorophyll *a* (Chl-*a*) is used as a proxy of phytoplankton biomass and is an important indicator in assessing the trophic status of freshwater ecosystems [3]. It is also widely used as a surrogate measure of the potential public health risk imposed by cyanobacterial blooms [4]. Routine monitoring of Chl-*a* concentrations in aquatic ecosystems is an important part of the environmental reporting requirements of many countries [5].

In the past two decades, substantial research has focused on the dynamics of Chl-*a* concentrations in aquatic ecosystems using in situ observations and remote sensing imagery [6,7]. These studies have demonstrated remarkable heterogeneity in the spatial and temporal distribution of Chl-*a* among various sampling sites and regions within a waterbody. Chl-*a* concentrations are dependent upon many physical (e.g., water temperature, light level, and wind regime), chemical (nutrient concentrations), and biological (algal growth rate and algal biomass) factors [8]. Currently, the overwhelming majority of lakes in China suffer from eutrophication and algal blooms, for which Chl-*a* is commonly used as a key indicator [9]. In shallow lakes with water depths <3 m, the variations in the spatiotemporal distribution of Chl-*a* in hypertrophic lakes may be different from those in mesotrophic or oligotrophic lakes [10]. These hypertrophic lakes tend to be dominated by surface blooms composed of buoyant cyanobacteria that are subject to rapid redistribution by wind and water movement [11].

Modeling is a useful approach for resolving interacting environmental processes and supporting management practices related to addressing lake eutrophication [12]. Several models have been introduced to simulate Chl-*a* concentrations. For example, the Chl-*a* concentrations in the coastal region of Hong Kong were fitted with a model using genetic programming [13] and an unsteady three-dimensional eutrophication model [14]; moreover, artificial neural networks have been used to simulate the Chl-*a* concentrations at a daily time scale in a lowland river in Germany [15]. The horizontal distribution of phytoplankton biomass in Lake Taihu, China, has been forecast over a short period using a two- and three-dimensional hydrodynamic-phytoplankton model [16,17]. The accuracy of the simulations of Chl-*a* concentrations in freshwater lakes has been progressively improved with increasingly sophisticated models [18]; however, it is still difficult for policy makers to use the models due to the high skill levels required for many of the models.

The relationships between Chl-*a* and nutrients have been identified using empirical models. For example, the correlation between Chl-*a* and total phosphorus (TP) has been studied, and this correlation has been used to calculate Chl-*a* concentrations [19]. Phosphorus is considered the primary limiting nutrient in freshwater lakes [20]; however, phytoplankton growth is also controlled by several other environmental factors, including light and water temperature (WT), as well as by other potentially limiting nutrients (e.g., nitrogen [21]) and trace elements (e.g., molybdenum [22]). The limiting factors usually change rapidly throughout the period of algal growth and differ among various regions of large freshwater lakes [23], and these variations can jointly influence the simulation accuracy of statistical models incorporating Chl-*a* and TP. Liu et al. [24] explored the effect of lake water chemistry on Chl-*a* concentrations in Lake Qilu using a multivariate linear regression (MLR), and found that the statistical model can rapidly and successfully simulate the variation in Chl-*a* concentrations. The MLR model has been easily used in supporting management decisions about eutrophication of Lake Qilu. In this study, the relationships between the Chl-*a* concentration and key environmental factors in Meiliang Bay were analyzed by MLR for the period from 2005 to 2015. The objectives of this study were to (1) illustrate the variation in the spatiotemporal distributions of Chl-*a* concentrations in Meiliang Bay, and (2) develop an MLR model to relate the Chl-*a* concentrations to key environmental factors that potentially regulate algal growth.

## 2. Materials and Methods

### 2.1. Study Area and Sampling Sites

Meiliang Bay, a northern bay of Lake Taihu in China, covering a surface area of 131 km^2^, has a mean water depth of 2.0 m and a water volume of 2.6 × 10^8^ m^3^. The Zhihu River and the Liangxi River flow into the bay (Figure 1). The prevailing wind direction from March to August is southeast, which promotes the aggregation of algal scums in the nearshore zone in northwestern Meiliang Bay. Four sampling sites (S1 to S4) were established in Meiliang Bay. S1 was in the north of Meiliang Bay, S2 was close to a major river water input, S3 was in the open water of the bay, and S4 was at the southern end between the bay and the main basin of the lake (Figure 1). Six other sampling sites (S5 to S10) were also established in the main basin of Lake Taihu. S5 was located in the region where cyanobacterial blooms frequently occur, S6 was in the area with frequent shipping traffic, S7 was in the zone with proximity to inflows, S8 was in the central region, and S9 and S10 were in submerged macrophyte- and aquaculture-dominated zones in Lake Taihu, respectively (Figure 1).

### 2.2. Water Sample Collection and Analysis

Water samples were collected monthly from January 2005 to December 2015 from four sites in Meiliang Bay and six additional sites in Lake Taihu. Two liters of mixed water samples at 10, 30, and 50 cm below the water surface were collected and transported on ice to the laboratory for further analysis. Three-hundred milliliters of water from each site was filtered through glass microfiber filters (nominal pore diameter 1.2 μm, GF/C, Whatman, United Kingdom). The filtrate was used to measure the concentrations of nitrate (NO_3_^−^-N), ammonium (NH_4_^+^-N), and dissolved organic matter (DOM). The concentrations of NO_3_^−^-N and NH_4_^+^-N were measured by a San^++^ Continuous Flow Analyzer (1100/1150 Sampler, Skalar, Netherlands). The raw water samples were used to determine the concentrations of total nitrogen (TN) and TP. Unfiltered sample water from each sampling site was digested using potassium persulfate before measuring the concentrations of TN and TP, and the filtrate of raw water was used to measure DOM according to the method of Jin and Tu [25]. The extraction of Chl-*a* from filters used for the detection of dissolved inorganic N was carried out with hot ethanol [26] and calculated according to the absorbance of the filtrates of each sampling site read at 665 and 750 nm using a spectrophotometer (UV-2101 PC, Shimadzu Co., Kyoto, Japan). WT, pH, and dissolved oxygen (DO) were monitored in situ at all sampling sites using a water quality sonde (YSI-6600V2, Yellow Spring Instruments, Cleveland, OH, USA) with calibrated sensors prior to use, and Secchi depth (SD) was measured in situ with a Secchi disk at each sampling site. Each nutrient and Chl-*a* measurement was made in triplicate.

### 2.3. Multivariate Statistical Methods

#### 2.3.1. Principal Component and Factor Analyses

The applicability of the principal component analysis (PCA) method to the analysis of water quality parameters at sampling sites S1 to S4 was first verified by Bartlett’s sphere test (χ^2^). The resulting χ^2^ value was 2813.3 (df = 36, *p* < 0.001), which indicated that the PCA was applicable to the analysis of the water quality dataset [27]. Principal components (PCs) with eigenvalues >1 were extracted and subjected to varimax rotation to generate varifactors (VFs). VFs reduce the contributions of variables of minor significance and provide information on the most meaningful parameters, and they can thus describe the whole dataset while allowing for data reduction with a minimal loss of original information [27].

#### 2.3.2. MLR Analysis

The distributions of ten environmental variables measured at S1 to S4 from January 2005 to December 2015 were examined for normality with a one-sample Kolmogorov–Smirnov (one-sample K–S) test. All of the *p*-values (two-tailed) were >0.05, indicating that the water quality variables had normal distributions. Prior to performing a multiple regression analysis, the collinearity among the independent variables should be diagnosed according to the eigenvalues, condition index (CI), variance proportion, tolerance, and variance inflation factor (VIF) [24]. Chl-*a* was used as the dependent variable in an MLR where partial regression coefficients for the independent variables were selected using a *t*-test at a significance level of 0.05 using SPSS 19.0 software (SPSS Inc., Chicago, IL, USA).

#### 2.3.3. MLR Model Performance Evaluation

The MLR model was validated using an independent data set from sampling sites S5 to S10 (Figure 1) from 2005 to 2015, with the coefficient of determination (*R*^2^), efficiency coefficient (E), and root-mean-squared error (RMSE) to observation standard deviation (STDEV) ratio (RSR).

*R*^2^ was calculated as follows: (1)$$R^{2}={\left\{\frac{\sum_{i=1}^{n}\left(o_{i}-o\right)\left(f_{i}-\bar{f}\right)}{\sqrt{\sum_{i=1}^{n}{\left(o_{i}-\bar{o}\right)}^{2}\sum_{i=1}^{n}{\left(f_{i}-\bar{f}\right)}^{2}}}\right\}}^{2}\;.$$

RMSE was calculated as: (2)$$\mathrm{RMSE}=\sqrt{\frac{\sum_{i}^{n}{\left(o_{i}-f_{i}\right)}^{2}}{n}}\;.$$

STDEV was calculated as: (3)$$\mathrm{STDEV}=\sqrt{\frac{\sum_{i}^{n}{\left(o_{i}-\bar{o}\right)}^{2}}{n}}\;.$$

E was calculated as: (4)$$E=1-\frac{\sum_{i=1}^{n}{\left(o_{i}-f_{i}\right)}^{2}}{\sum_{i=1}^{n}{\left(o_{i}-\bar{o}\right)}^{2}}\;.$$

RSR was calculated as: (5)$$\mathrm{RSR}=\frac{\mathrm{RMSE}}{\mathrm{STDEV}}=\frac{\sqrt{\sum_{i=1}^{n}{\left(o_{i}-f_{i}\right)}^{2}/n}}{\sqrt{\sum_{i=1}^{n}{\left(o_{i}-\bar{o}\right)}^{2}/n}}=\frac{\sqrt{\sum_{i=1}^{n}{\left(o_{i}-f_{i}\right)}^{2}}}{\sqrt{\sum_{i=1}^{n}{\left(o_{i}-\bar{o}\right)}^{2}}}\;.$$
where $o_{i}$ and $f_{i}$ are the *i*th observed and simulated Chl-*a* concentrations, respectively; $\bar{o}$ and $\bar{f}$ denote the mean observed and simulated Chl-*a* concentrations, respectively; and *n* is the number of data points for each sampling site (S5 to S10, *n* = 132) from 2005 to 2015, as well as for six sampling sites overall (*n* = 72) in 2005, 2010, and 2015. *R*^2^, which ranges from zero to one, represents the percentage of variation explained, whereas *R*^2^ >0.5 is considered acceptable [4]. E = 1 is indicative of a perfect match of the fitted values to the observed values, E = 0 denotes that the simulated values are similar to the observed values, and E < 0 indicates an unacceptable model simulation performance [28]. The RSR ranges from zero, indicating perfect model simulation performance, to any positive value, with satisfactory model performance at RSR <0.70 [29].

The significance of the differences in the average seasonal and annual Chl-*a* concentrations among stations (S1 to S4) was tested by one-way analysis of variance (ANOVA) in conjunction with Tukey’s test for multiple comparisons, and correlations with the different water quality variables were determined by a Pearson correlation coefficient analysis.

## 3. Results

### 3.1. Spatiotemporal Variation in Chl-a Concentration in Meiliang Bay

From January 2005 to December 2015, the monthly Chl-*a* levels varied between 2.6 and 330.0 μg/L at S1, 2.8 and 221.0 μg/L at S2, 4.1 and 235.0 μg/L at S3, and 7.0 and 161.0 μg/L at S4. The monthly Chl-*a* levels peaked in May 2007 at S1, in November 2007 at S2, in November 2009 at S3, and in October 2013 at S4 (Figure 2). The 11 year monthly average Chl-*a* at S1 (45.2 ± 49.1 μg/L) was significantly higher than that at S2 (35.6 ± 36.4 μg/L, *p* < 0.01), S3 (38.2 ± 40.3 μg/L, *p* < 0.01), and S4 (35.6 ± 32.7 μg/L, *p* < 0.01). Variations in the mean Chl-*a* levels for spring (March to May), autumn (September to November), and winter (December to February) were low among the four sampling sites but the summer (June to August) values at S1 were significantly higher than those at S3 and S4 (Figure 3). Chl-*a* was not significantly different between spring and winter, except at S1, although the values in these months were significantly lower than those in summer and autumn (Figure 3). The annual mean Chl-*a* concentrations varied from 24.7 to 105.9 μg/L at S1, 19.3 to 55.7 μg/L at S2, 20.9 to 52.0 μg/L at S3, and 21.7 to 45.4 μg/L at S4, with moderately high coefficients of variation of 49.3%, 36.7%, 26.3%, and 22.3% at S1, S2, S3, and S4, respectively. The annual mean Chl-*a* levels increased from 2005 to 2007, decreased from 2007 to 2012, and increased from 2012 to 2015 at S1 and S2 (Figure 4). At S3 and S4, the annual mean Chl-*a* declined from 2005 to 2007, increased from 2007 to 2009, decreased from 2009 to 2012, and increased from 2012 to 2015 (Figure 4).

### 3.2. MLR Model of Chl-a Concentrations and Key Environmental Factors

Significant positive linear correlations (*p* < 0.01) occurred between the Chl-*a* concentration and TP, DOM, WT, and pH for the four Meiliang Bay sampling sites collectively (*n* = 528), and the Pearson correlation coefficients were 0.57 for TP, 0.73 for DOM, 0.37 for WT, and 0.44 for pH. Significantly negative correlations (*p* < 0.01) were also observed for Secchi depth (ln(SD)) (*r* = −0.11), DO (*r* = −0.12), and NO_3_^−^-N (*r* = −0.28); however, the correlations were not significant for TN and NH_4_^+^-N (Figure 5).

Three PCs with eigenvalues exceeding 1.0 were extracted from the nine water quality variables using the PCA method with varimax rotation (Table 1). PC1, accounting for 25.9% of the total variance, showed positive correlations with DOM and TP. PC2, explaining 25.3% of the total variance, was positively correlated with NH_4_^+^-N and TN and negatively correlated with pH. PC3, accounting for 21.3% of the total variance, was positively correlated with WT and negatively correlated with DO.

On the basis of the significant correlations between the water quality variables and Chl-*a* concentrations, as well as the major contributions of the water quality variables to three PCs, five water quality variables (TP, DOM, WT, pH, and DO) were screened to develop an MLR model over all four stations in Meilang Bay. The eigenvalues, CI, variance proportion, tolerance, and VIF, which are characteristic parameters of the collinearity diagnostics, are listed in the Appendix A and Table 2, indicating that there was collinearity between the constant and DO contents and, thus, DO was excluded from the five water quality variables for the development of the MLR model. The non-standardized partial regression coefficients of the four independent variables are provided in Table 2. The MLR model was as follows:

Chl-*a* = −129.84 + 1.80WT + 12.15DOM + 122.34TP + 6.14pH.(6)

The *R*, *F*, and *p* values were 0.799, 230.7, and <0.001, respectively, which indicate that the MLR model successfully described the relationships between Chl-*a* concentration and the four independent water quality variables (WT, DOM, TP, and pH) from Meiliang Bay from 2005 to 2015. Subsequently, the MLR was used to simulate the dynamics of the Chl-*a* concentration at each sampling site (from S5 to S10) during 2005 and 2015, as well as at six sampling sites overall in 2005, 2010, and 2015. The simulated Chl-*a* concentrations were significantly correlated with the observed values (Figure 6), as indicated by the *R*^2^, E, and RSR values that ranged from 0.52 to 0.89, 0.08 to 0.76, and 0.44 to 0.69 (Table 3), respectively. These results indicate that the Chl-*a* concentrations simulated by the MLR model are reliable at the six other sites in Lake Taihu. Additionally, the percentage of the standardized residuals between the simulated and observed Chl-*a* concentrations ranging from −2.0 to 2.0 was 94.7% for S5, 94.0% for S6 and S7, 96.2% for S8, 95.4% for S9 and S10, 93.1% for 2005, 94.4% for 2010, and 97.2% for 2015 (Figure 7), suggesting that the MLR model simulation performance was acceptable and that the model could successfully simulate the variation in the Chl-*a* concentration in different areas of very eutrophic lakes.

## 4. Discussion

### 4.1. Factors Influencing the Distribution of Chl-a Concentration in Meiliang Bay

Nitrogen and phosphorus play an important role in the spatiotemporal distributions of the Chl-*a* concentrations in aquatic environments [30]. Furthermore, the TN to TP ratio (NPR) is commonly regarded as an indicator of the likelihood that either phosphorus, nitrogen, or either nutrient could limit phytoplankton growth [31]. The NPRs in Meiliang Bay ranged from 4.7 to 190.0, with a mean of 36.2. The percentage of the NPR exceeding 16, which is the mass ratio required to meet the physiological demands of phytoplankton for nitrogen and phosphorus concentrations [32], was significantly higher in Meiliang Bay in summer and autumn compared with spring and winter (Figure 8), which indicates that phytoplankton growth in summer and autumn is potentially predominantly limited by the phosphorus supply [33]. Because optimal NPRs vary among phytoplankton species, it is difficult for NPRs to identify a limiting nutrient for a multi-species community [34]. However, *Microcystis aeruginosa* is a dominant species in Meiliang Bay [35]. Moreover, a significant correlation between the Chl-*a* and TP in Meiliang Bay suggests that phytoplankton biomass increases with increasing phosphorus concentration and that phosphorus may be a key factor limiting phytoplankton growth, as demonstrated by Xu et al. [36]. The absence or weakness of relationships between Chl-*a* and the TN, NO_3_^−^-N, and NH_4_^+^-N concentrations in Meiliang Bay indicate that nitrogen likely has a lesser influence on phytoplankton growth and it might generally be at sufficient levels to meet nutritional demands, which is consistent with the results of previous studies [37,38].

DOM primarily constitutes humic substances, polysaccharides, and proteins [39], whose breakdown serves as a potential source of nutrient supply for phytoplankton growth [40]. DOM can be released from the pore water of subsurface sediments into overlying water due to hydrodynamic disturbance [41], and derives from phytoplankton, including the exudation of live phytoplankton and degradation of dead phytoplankton [42], as well as allochthonous inputs from catchments [43]. The significant relationship between DOM and Chl-*a* in Meiliang Bay implies that DOM might play an important role in providing nutrition for phytoplankton growth, although the contribution of different DOM sources to algal growth was not identified. This finding is consistent with Ye et al. [44], who demonstrated that allochthonous sources have a greater influence on the origin of DOM concentrations in Lake Taihu than autochthonous sources using an isotope tracing technique.

Spatiotemporal variations in Chl-*a* concentration are also impacted by meteorological factors [45]. The warming of surface water accelerates the growth rates of bloom-forming cyanobacteria in particular, causing them to more rapidly reach their maximal growth rates and increasing the frequency, intensity, and duration of cyanobacterial blooms in eutrophic freshwater lakes [8,46]. In Meiliang Bay, WT was significantly positively correlated with Chl-*a* concentration (Figure 5), which was most likely because WT accelerated cyanobacteria growth in particular. The Chl-*a* concentrations at S1 in summer and autumn were much higher than those at the three other sampling sites. In summer and autumn, the prevailing wind is from the east–southeast, which promotes the accumulation of cyanobacteria in the north of Meilang Bay where S1 was located, as evidenced by the rapid increases in Chl-*a* in the north of Meilang Bay [47].

S2 was situated in a predominantly downwind zone of Meiliang Bay; however, the large inflow from Zhihu River is near S2. The mean inflow ranged from 2.1 to 360 m^3^/s in summer (June to August) during 2005 and 2015, and this condition might effectively suppress the accumulation of buoyant cyanobacteria in the upper water layer. Li et al. [48] carried out field enclosure experiments on the effect of flow velocities (from 0.03 to 0.3 m/s) on phytoplankton biomass and found that the Chl-*a* concentrations in the upper water layer were significantly negatively related to flow rates. Therefore, we speculated that the lake currents caused by the large inflow from the Zhihu River might contribute to the vertical mixing and/or flushing of cyanobacteria within the water column and subsequently reduce the Chl-*a* concentrations in the upper water layer. Furthermore, incoming and outgoing fishing vessels and cargo ships near S2 result in water disturbances that likely also prevent the high aggregation of phytoplankton biomass in the upper water layer. The predominant wind-driven currents at S3 and S4, which were located in the open water of Meiliang Bay, would likely be higher and less favorable for the massive accumulations of cyanobacteria, which were reflected by the decreased Chl-*a* concentrations, as noted in other studies [47].

### 4.2. Performance of the MLR Model for Chl-a Concentrations in Lake Taihu

The relationships between Chl-*a* concentrations and physicochemical factors in aquatic ecosystems have been studied using multivariate statistical techniques [49]; however, regression models derived from the literature that relate Chl-*a* concentrations to water quality parameters differ due to variations in the trophic status, WT, water color, and mixing regime [50]. In our study, the MLR model demonstrated that it was reliable and acceptable for simulating the Chl-*a* concentrations derived from stations S5 to S10 in Lake Taihu from January 2005 to December 2015 (Figure 6). Some discrepancies between the model performance with the training data set (S1–S4) and the validation data set (S5–S10) may be attributable to differences in the trophic status of different zones in Lake Taihu. The trophic statuses at S5 to S8 belong to eutrophic levels, similar to those of S1 to S4. S9 and S10 have a much lower trophic status. S9 was located in a submerged macrophyte-dominated region comprising mostly *Potamogeton maackianus* A. Bennett, where the mean Chl-*a* concentrations were low (mean = 10.5 µg/L). Aquatic plants, especially submerged macrophytes, compete with phytoplankton for nutrients [51] and can excrete allelopathic substances that inhibit algal growth [52]. S10 was located in an aquaculture-dominated enclosed zone where the dominant submerged macrophyte is *Potamogeton malaianus* Miq., with areal cover >90% [53]. This area is heavily artificially manipulated to provide shelter for crabs. Because of a large-scale enclosure net at this site, wind speeds are decreased by 75% relative to the values over open water [54]. The resulting reduction in wind disturbance of the sediments reduces the release of phosphorus and nitrogen from the sediment into the overlying water in this enclosure-dominated region [55]. These observations indicate that aquatic vegetation and enclosure aquaculture cause different trophic statuses and likely result in reduced goodness of fit of the model for these validation sites (S9 and S10) compared with the other validation sites (S5 to S8) in Lake Taihu (Table 3).

A 38-parameter three-dimensional ecological model that integrated the hydrodynamic, chemical, and biological processes of nutrients was developed to simulate and predict the spatiotemporal variations in algal blooms in Lake Taihu [16]. Although the acceptable average annual relative deviation between the simulated and observed Chl-*a* concentrations from different sampling sites in Meiliang Bay was less than 40%, high monthly relative deviations that varied from 0.0% to 90.8% were observed. Huang et al. [56] developed a hydrodynamic phytoplankton model to simulate the spatiotemporal distribution of phytoplankton biomass in Lake Taihu. An accuracy of 78.7% between the simulated and observed Chl-*a* concentrations was achieved. However, the mean percent error (13.4%) and mean absolute percent error (58.2%) indicated that further improvements, for example, by reducing the uncertainty of the model inputs and/or by improving the parameter calibration, should be made. Jiang et al. [57] simulated the Chl-*a* concentrations in different subareas of Lake Taihu using a 40-parameter environmental fluid dynamics code (EFDC) model and found that the fitted Chl-*a* concentrations were basically consistent with field observations. The analyses of the sensitivity and uncertainty of the simulations imply that the observed Chl-*a* concentrations, water temperature, light, and simulation time are primary factors influencing the EFDC simulation accuracy. These complicated models can demonstrate the mechanism of the rapid changes in both the spatial and temporal heterogeneity of phytoplankton biomass; however, compared with the MLR model in our study, they usually require more data for input and calibration, which are rarely available at adequate temporal and spatial resolution. This lack of field data is a major challenge for the full validation of the models.

### 4.3. Implications for Lake Management

Although some measures have been taken to prevent exogenous nutrients from entering water bodies and reduce internal nitrogen and phosphorus release from sediments, the control of eutrophication and nuisance cyanobacterial blooms in aquatic ecosystems, particularly in shallow lakes, remains a global challenge. Because of an incomplete understanding of the mechanisms of cyanobacterial bloom formation and the nutrient levels and water temperatures necessary for the onset of algal blooms, strategies for mitigating harmful cyanobacterial blooms in shallow eutrophic water bodies, for example, in Lake Taihu, China, and Lake Okeechobee, USA [58], are not always effective [59]. To optimize the measures for controlling eutrophication and algal blooms and decrease operational expenditures, the quantitative links between key driving factors and the phytoplankton biomass (Chl-*a* concentration) response should be identified. Although discrepancies were observed between the measured Chl-*a* concentrations and the simulated values due to different ecological zones with various trophic states, submerged vegetation cover, and anthropogenic activities, the MLR model developed in the present study successfully simulated the dynamics of Chl-*a* concentrations in multiple ecological zones in Lake Taihu (Figure 6). Therefore, when the MLR model is applied to other eutrophic lakes similar to Lake Taihu, other factors, such as aquatic vegetation and aquaculture, should be considered to further improve the simulation accuracy.

Policy makers and managers emphasize operationally feasible management strategies to limit rapid increases in phytoplankton biomass, especially bloom-forming cyanobacteria, which requires screening the regulated key driving factors that influence the spatiotemporal distribution of phytoplankton biomass (Chl-*a* concentration). Compared with complicated two- and three-dimensional mechanisms and process models that frequently require extensive data for model calibration and validation and a high level of user skill, statistical regression models can provide an alternative to eutrophication control and harmful cyanobacterial bloom suppression because of reduced data and user skill requirements. Several in situ observatory platforms in Lake Taihu have been constructed to monitor the dynamics of water quality in real-time, and many routine parameters, such as TP, WT, DOM, and pH, are easily obtained via these platforms, although it is not as easy to regulate these parameters with artificial methods (e.g., the restoration of submerged macrophytes). The MLR model that integrated the Chl-*a* concentration with nutrient and meteorological factors of interest can effectively assist government officials in making decisions associated with the control of phytoplankton biomass in eutrophic lakes [24]. Nutrient loadings (TP and/or DOM concentrations) can be artificially regulated relatively easily by reducing exogenous inputs from the catchment and inhibiting endogenous releases from sediments into overlying water.

As discussed above, the wind regime, submerged macrophytes, and human activities, such as enclosure aquaculture, are additional factors likely affecting the spatiotemporal distribution of Chl-*a* concentrations [60]. Due to a shortage of long-term monitoring data, the interactive effects of these factors on changes in Chl-*a* concentrations in phytoplankton-dominated and submerged-macrophyte-dominated zones in Lake Taihu were not incorporated into the MLR model. It is difficult to simultaneously monitor water quality, ecological indicators, and hydrometeorological factors in situ; however, we appeal to lake managers and local authorities for more capital inputs related to observatory platform construction in large eutrophic lakes to assist with the validation of models. In combination with information on the dynamics of hydrometeorological factors (wind-driven lake current and wave height) and anthropogenic activity (aquatic vegetation cover), the MLR model can play an increasing role in making decisions associated with eutrophication and phytoplankton biomass control in freshwater ecosystems.

## 5. Conclusions

In this study, the spatiotemporal dynamics of Chl-*a* concentrations were investigated, and a multiple linear regression model was developed using 11 year water quality data from Meiliang Bay in Lake Taihu. Our study demonstrated the heterogeneity of Chl-*a*, with the northern near-shore zone presenting significantly higher concentrations than the other zones of Meiliang Bay. The Chl-*a* concentrations in Meiliang Bay fluctuate extensively on an annual time scale and are higher in summer and autumn than in spring and winter, and these fluctuations are primarily driven by fluctuating nutrients (e.g., TP) and meteorological factors (e.g., WT). The interactive effects of the key environmental factors on the spatiotemporal dynamics of Chl-*a* concentrations in Meiliang Bay can be characterized by a MLR model that incorporates TP, DOM, pH, and WT. The MLR model successfully simulates the variations in Chl-*a* concentrations in other regions of Lake Taihu, which gives local authorities an alternative to make decisions related to taking physical, chemical, and biological measures (e.g., increasing submerged macrophyte coverage, vertical mixing, and horizontal flushing of water bodies) to decrease nutrient loadings and/or suppress aggregations of phytoplankton biomass, particularly near drinking water resources.

## Figures and Tables

**Figure 1 ijerph-16-04553-f001:**
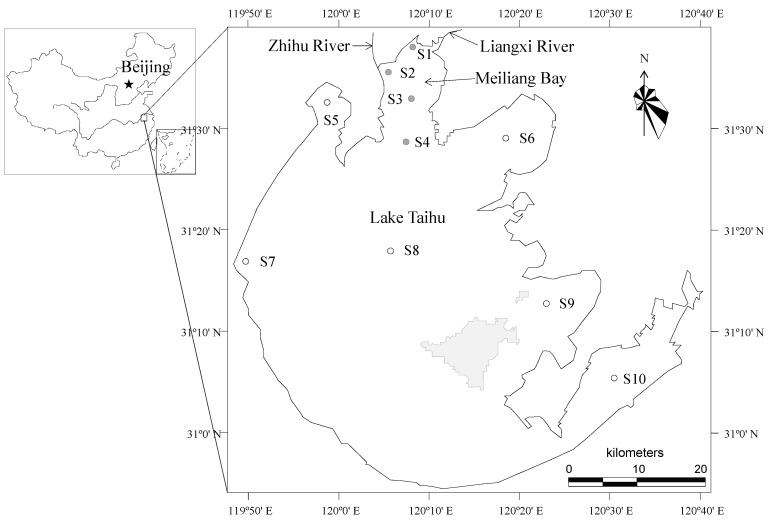
Distribution of the sampling sites established for analyzing the spatial and temporal variations in the Chlorophyll *a* (Chl-*a*) level, constructing a multiple regression model (S1 to S4) and validating the multiple regression model (S5 to S10). Filled circles (●) are the observed sites for the analysis of the spatial and temporal variations in Chl-*a* levels and construction of a multiple regression model, and open circles (○) are the observed sites for the validation of the multiple regression model.

**Figure 2 ijerph-16-04553-f002:**
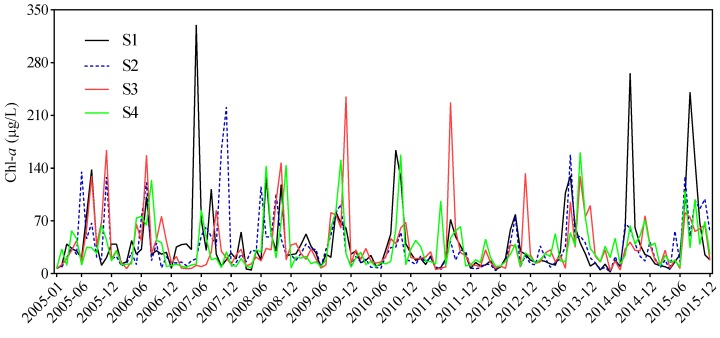
Monthly variation in Chl-*a* concentration at the four sampling sites in Meiliang Bay during 2005 and 2015.

**Figure 3 ijerph-16-04553-f003:**
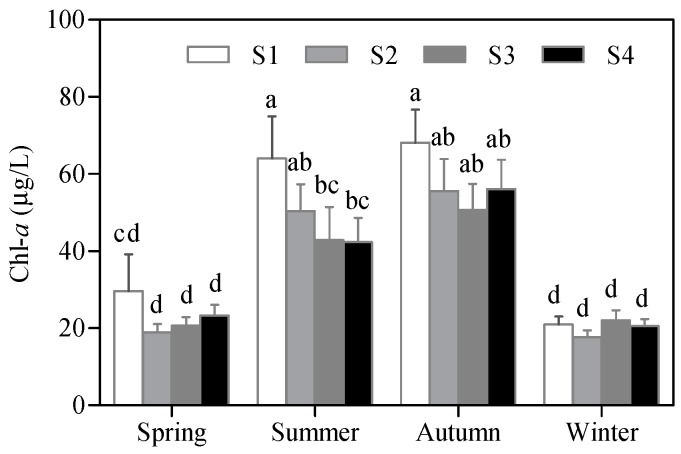
Seasonal variation in the Chl-*a* concentration at the four sampling sites in Meiliang Bay during 2005 and 2015. All data are shown as the mean ± standard deviation. Chl-*a* concentrations with the same letter do not differ significantly among the sampling sites within a season (Tukey test, *p* ≤ 0.05).

**Figure 4 ijerph-16-04553-f004:**
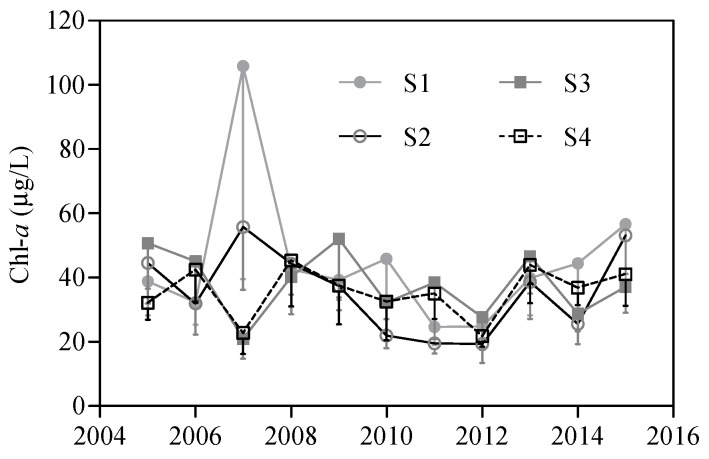
Annual variation in Chl-*a* concentration at the four sampling sites in Meiliang Bay during 2005 and 2015. All data are shown as the mean ± standard deviation.

**Figure 5 ijerph-16-04553-f005:**
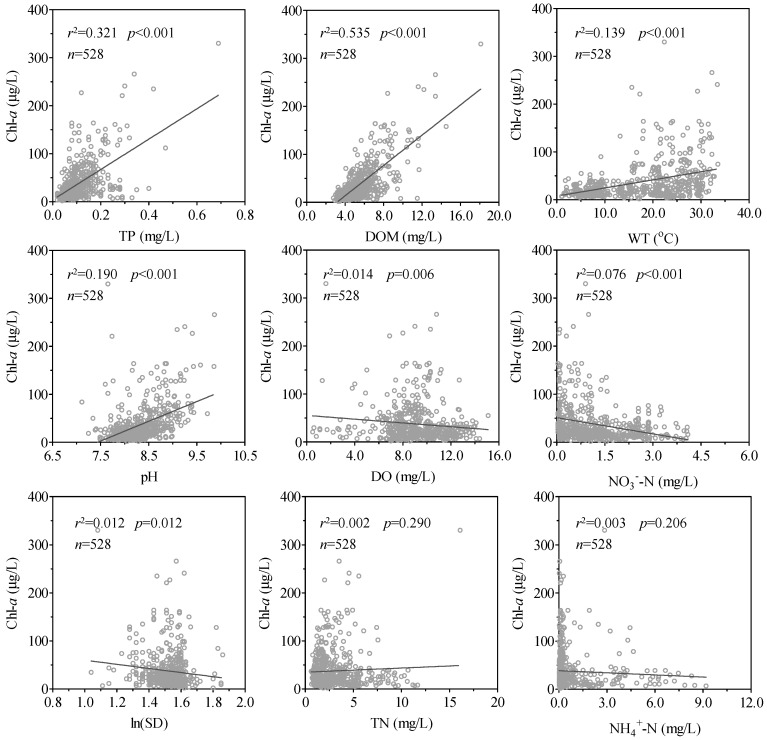
Correlations between Chl-*a* concentration and the nine water quality variables at the four sampling sites in Meiliang Bay. TP: total phosphorus; DOM: dissolved organic matter; WT: water temperature; DO: dissolved oxygen; NO_3_^−^-N: nitrate; TN: total nitrogen; NH_4_^+^-N: ammonia; SD: transparency.

**Figure 6 ijerph-16-04553-f006:**
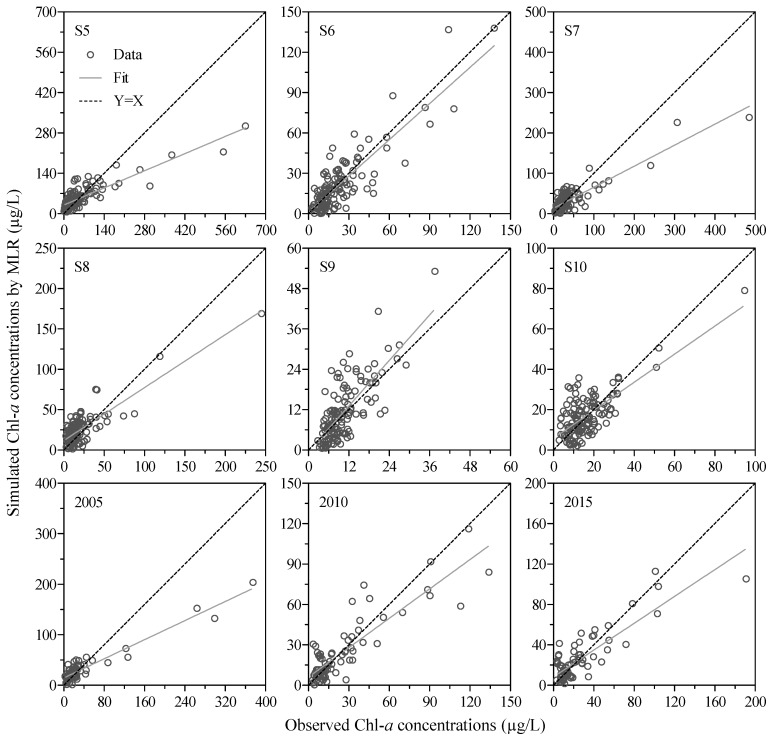
Correlations between the simulated and observed Chl-*a* concentrations at each sampling site (from S5 to S10) during 2005 and 2015 and at six sampling sites overall in 2005, 2010, and 2015, respectively, in Lake Taihu.

**Figure 7 ijerph-16-04553-f007:**
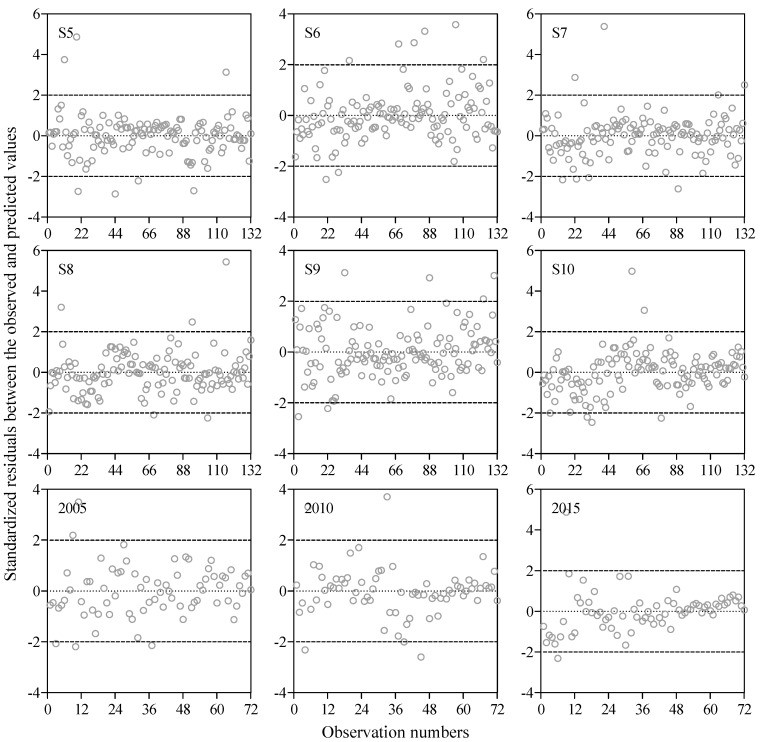
Standardized residuals between the simulated and observed Chl-*a* concentrations at each sampling site (from S5 to S10) during 2005 and 2015 and at six sampling sites overall in 2005, 2010, and 2015 in Lake Taihu.

**Figure 8 ijerph-16-04553-f008:**
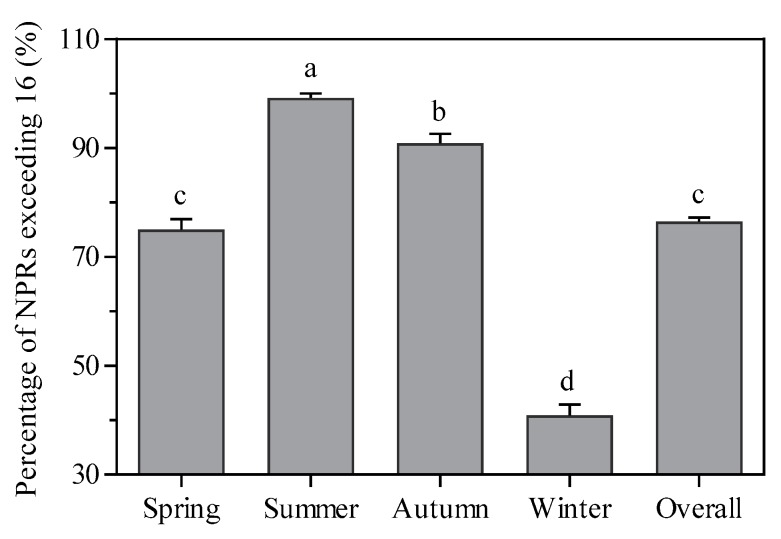
Percentages of mass concentration ratios of total nitrogen to total phosphorus (NPRs) exceeding 16 in Meiliang Bay among the different seasons. All data are shown as the mean ± standard deviation. The percentages of NPRs exceeding 16 with the same letter do not differ significantly among the different seasons (Tukey test, *p* ≤ 0.05).

**Table 1 ijerph-16-04553-t001:** Results of the principal components with varimax rotation based on the water quality dataset from 2005 to 2015 in Meiliang Bay. Bold values denote strong loadings. PC: principal component.

Variable	PC1	PC2	PC3	Communality
WT	0.028	−0.332	**0.810**	0.768
SD	−0.487	−0.110	0.136	0.268
DO	−0.139	−0.333	**−0.894**	0.930
pH	0.137	**−0.862**	0.036	0.763
NO_3_^−^-N	−0.065	0.477	−0.463	0.445
NH_4_^+^-N	0.545	**0.703**	0.098	0.801
TN	0.609	**0.720**	−0.083	0.896
DOM	**0.863**	−0.221	0.235	0.849
TP	**0.799**	0.088	0.398	0.805
Eigenvalue	3.048	2.341	1.137	
% Total variance	25.904	25.257	21.336	

**Table 2 ijerph-16-04553-t002:** Multiple regression results for the water quality parameters and the results of collinearity tests. VIF: variance inflation factor.

Variable	Unstandardized Coefficients		*t*-test	*p*	Collinearity Statistics	
	B	Standard Error			Tolerance	VIF
Constant	−129.84	8.19	−15.86	0.00		
WT (°C)	1.80	0.15	11.64	0.00	0.60	1.68
TP (mg/L)	122.34	24.90	4.91	0.00	0.35	2.88
DOM (mg/L)	12.15	0.89	13.69	0.00	0.41	2.45
pH	6.14	0.63	9.82	0.00	0.49	2.04

**Table 3 ijerph-16-04553-t003:** The coefficients of the linear equations coupled with the simulated concentrations (C_s_) and the observed Chl-*a* concentrations (C_o_) at each sampling site (from S5 to S10) and the six sampling sites overall in 2005, 2010, and 2015 and the corresponding parameters used to assess the goodness of fit to Chl-*a*. ^**^ Significant value at the 0.01 level.

	C_s_ = aC_o_ + b		*n*	Parameters Assessing Goodness of Fit			*p*
	a	b		*R* ^2^	E	RSR	
S5	0.4244	30.094	132	0.7326	0.6026	0.5333	<0.001 ^**^
S6	0.8965	1.2164	132	0.7405	0.7048	0.4774	<0.001 ^**^
S7	0.5209	13.488	132	0.7865	0.6931	0.4846	<0.001 ^**^
S8	0.6445	12.029	132	0.6792	0.6349	0.5172	<0.001 ^**^
S9	1.1267	−0.3766	132	0.5655	0.0787	0.6925	<0.001 ^**^
S10	0.6227	6.7797	132	0.5179	0.4040	0.6111	<0.001 ^**^
2005	0.4729	14.712	72	0.8894	0.6911	0.4858	<0.001 ^**^
2010	0.7279	6.1363	72	0.7622	0.7599	0.4400	<0.001 ^**^
2015	0.6646	8.3166	72	0.7236	0.7199	0.4678	<0.001 ^**^

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
