# Peer review of "Variations in the Distribution of Chl-a and Simulation Using a Multiple Regression Model"

_ijerph, 2019, doi:10.3390/ijerph16224553_

Round 1
Reviewer 1 Report
General Comments
The manuscript is a nice contribution to the field of limnological modelling and eutrophication control. The authors based on the multiple regression method simulated the chlorophyll’s spatiotemporal dynamics in Lake Taihu and produced good modelling results. This manuscript is good quality work; however, some minor/moderate modifications are recommended.
Specific Comments
Lines 26-28: please make sure that the negative sign is placed/used according with your findings
Line 71: please change the words order “high skill levels” to “high level skills”
Add in the introduction part few studies, that are using MLR method to simulate the clorophyll parameter successfully; also, briefly report in this part the significance/advantages of MLR method as a modelling tool.
Line 104: the filled circles are black, while in Figure 1 are grey shaded. Please change in order to match.
Lines 142-145: please add reference
Lines 153-157: the equation number is given twice, please remove the extra
Line 191: the color shades that are used are making Figure 2 hard to be observed especially for S1, please use different shades.
Line 203: please add the word statistically before the word significant
Line 228: please justify why the DO parameter was excluded.
Author Response
ID: ijerph-626574
Title: Variations in the Distribution of Chl-a and Simulation Using a Multiple Regression Model
Reviewer 1:
We really appreciate the insightful comments provided by the reviewers, which have helped us to significantly improve the quality of our manuscript. As per the professional comments and questions, the manuscript has been checked carefully and revised accordingly (using “Track Changes”). Specifically, the main modifications and point-to-point responses were listed below corresponding to the comments.
General comments:
The manuscript is a nice contribution to the field of limnological modelling and eutrophication control. The authors based on the multiple regression method simulated the chlorophyll’s spatiotemporal dynamics in Lake Taihu and produced good modelling results. This manuscript is good quality work; however, some minor/moderate modifications are recommended.
Specific comments:
Lines 26-28: please make sure that the negative sign is placed/used according with your findings.Response: Thanks for your comments. We accepted your suggestion.
Line 71: please change the words order “high skill levels” to “high level skills”.Response: Thanks for your comments. We accepted this revision.
Add in the introduction part few studies that are using MLR method to simulate the clorophyll parameter successfully; also, briefly report in this part the significance/advantages of MLR method as a modelling tool.Response: Thanks for your valuable comments. In the introduction part, the relevant reports on a MRL model being successfully used to simulate variation of Chl-a concentration in a eutrophic lake was added.
Line 104: the filled circles are black, while in Figure 1 are grey shaded. Please change in order to match.Response: Thanks for your comments. We accepted your suggestion.
Lines 142-145: please add referenceResponse: Thanks for your comments. We accepted your suggestion.
Lines 153-157: the equation number is given twice, please remove the extra.Response: Thanks for your suggestion. In the revised manuscript, we carefully checked equation number sequences, and removed extra numbers in those equations.
Line 191: the color shades that are used are making Figure 2 hard to be observed especially for S1, please use different shades.Response: Thanks for your suggestion. We added different color for various legends in Figure 2. It seems to be clear to see.
Line 203: please add the word statistically before the word significant.Response: Thanks for your comments. We accepted your suggestion.
Line 228: please justify why the DO parameter was excluded.Response: Thanks for your valuable comment. The attachment of Supplementary Materials was not be uploaded during submitting the manuscript, which caused the explanation of DO exclusion to be insufficient and confusing. In the revised manuscript, we rewrote “The eigenvalues, CI, variance proportion, tolerance and VIF, which are characteristic parameters of the collinearity diagnostics listed in Table 2 and Supporting Information, were used to determine the five water quality variables for the development of the MLR model; however, DO was excluded.” to “The eigenvalues, CI, variance proportion, tolerance and VIF, which are characteristic parameters of the collinearity diagnostics, were listed in the Supporting Information and Table 2, indicating that there was the collinearity between the constant and DO contents and thus DO was excluded from the five water quality variables for the development of the MLR model.” Additionally, the Supplementary materials were also provided.
Supplementary Materials
Collinearity statistics of the five water quality variables derived from a multiple linear regression using a stepwise method.
|
Model |
D |
Eigenvalue |
CI |
Variance Proportions |
|||||
|
Constant |
COD |
DO |
TP |
WT |
pH |
||||
|
1 |
1 |
1.948 |
1.000 |
.03 |
.03 |
|
|
|
|
|
2 |
.052 |
6.108 |
.97 |
.97 |
|
|
|
|
|
|
2 |
1 |
2.932 |
1.000 |
.00 |
.01 |
.00 |
|
|
|
|
2 |
.067 |
6.613 |
.01 |
.95 |
.01 |
|
|
|
|
|
3 |
.001 |
48.579 |
.99 |
.04 |
.99 |
|
|
|
|
|
3 |
1 |
3.739 |
1.000 |
.00 |
.00 |
.00 |
.01 |
|
|
|
2 |
.228 |
4.048 |
.00 |
.00 |
.00 |
.34 |
|
|
|
|
3 |
.031 |
10.914 |
.01 |
.86 |
.00 |
.55 |
|
|
|
|
4 |
.001 |
57.891 |
.99 |
.14 |
1.00 |
.10 |
|
|
|
|
4 |
1 |
4.585 |
1.000 |
.00 |
.00 |
.00 |
.00 |
.01 |
|
|
2 |
.241 |
4.366 |
.00 |
.00 |
.00 |
.32 |
.07 |
|
|
|
3 |
.142 |
5.685 |
.00 |
.01 |
.00 |
.01 |
.79 |
|
|
|
4 |
.031 |
12.101 |
.01 |
.84 |
.00 |
.52 |
.00 |
|
|
|
5 |
.001 |
69.079 |
.99 |
.14 |
1.00 |
.15 |
.14 |
|
|
|
5 |
1 |
5.446 |
1.000 |
.00 |
.00 |
.00 |
.00 |
.00 |
.00 |
|
2 |
.321 |
4.117 |
.00 |
.00 |
.00 |
.17 |
.01 |
.02 |
|
|
3 |
.187 |
5.395 |
.00 |
.01 |
.00 |
.08 |
.30 |
.01 |
|
|
4 |
.032 |
13.098 |
.00 |
.87 |
.00 |
.55 |
.00 |
.01 |
|
|
5 |
.013 |
20.307 |
.05 |
.01 |
.01 |
.16 |
.39 |
.78 |
|
|
6 |
.001 |
82.817 |
.95 |
.10 |
.99 |
.04 |
.30 |
.18 |
|
D denotes dimension; CI means the condition index

Reviewer 2 Report
Deng et al. developed statistical multiple regression model to understand temporal and spatial variation of concentrations of Chlorophyll-a (Chla) in Lake Taihu. They use an impressive 11-year-long data series of monthly measurements from 10 stations. They identified water temperature (WT), dissolved organic matter (DOM), total phosphorous (TP) and pH to positively correlate with Chla, and dissolved oxygen (DO), nitrate and transparency to correlate negatively. In the model they used only those variables that correlated positively. The model was developed based on data series collected at some station and validated against the other stations. The outcome of validation indicates that the model can be used for management of Lake Taihu.
The study is not novel from the ecological point of view, but I imagine that it can be useful for monitoring and management purposes, thus I think it deserves publication in IJERPH. The manuscript is well written and nice to read. The Discussion can be shorten by removing or substantially shortening section 4.1 (on factor influencing Chla concentration), as this part is not novel at all. Concerning the model, I do not understand why the variables that correlate negatively were not included in the model. If the authors calculated such a model and it turned out to be worse that the current one, it should be included in the ms, otherwise it should be explained. Finally, I also think it would be valuable to validate the model against subsequent seasonal data if they are available (years 2016-2017/2018?).
Minor comments
L28: explain the abbreviation for SD
L111: what was the volume? This sentence is not very clear at the first reading
L113: DOM is defined as < 0.45 µm, thus these samples were collected incorrectly. Consider removing the DOM data from the model (see also below for comments to L291-301)
L118: explain the abbreviation for TP
L118-119: This sentence is a bit confusing with regard to making unclear whether the samples for DOM where filtered or not?
L125: how often was the sonde calibrated?
L148: Would it be possible to test the model on data from subsequent years (2016+)?
L170-L172: I do not understand why these correlations were calculated if you did the regression. Please explain.
L180-L181: please write how you defined seasons. Did you mean astronomical seasons (winter: 22 Dec-21 Mar; spring 22 Mar-20 Jun; summer: 21 Jun-21 Sep; autumn: 22 Sep-21 Dec) or meteorological seasons (winter: Dec-Feb; spring Mar-May; summer: Jun-Aug; autumn: Sep-Nov).
Figure 4: it seems that there was one outlier (Station S1, 2007), have you tested its influence on correlations, regression and the model performance?
L222-224: please explain why the variables that correlated negatively (mainly nitrogen) were not included in the model
L288: please explain why DO was excluded
Figure 8: please explain in the methods how the probabilities were calculated and present these results also in the result section
L291-301: This paragraph should more discuss that phytoplankton also produces and releases DOM, even if it’s not major DOM source in the Lake Taihu. This maybe the reason for the observed strong correlation (r2>0.5). Moreover, as mentioned before, samples for DOM measurements were not correctly collected in my opinion. This need to be critically address or DOM should be remove from the model
L324: The lake Taihu seems to be very shallow and large, so I imagine that it easily mixes even during summer. How likely are the occurrences of thermal stratification and how long does it last?
L354-355: “The trophic status at S5 to S8 is somewhat similar to that of S1 to S4” Could you be more specific here
Author Response
ID: ijerph-626574
Title: Variations in the Distribution of Chl-a and Simulation Using a Multiple Regression Model
Reviewer 2:
We really appreciate the insightful comments provided by the reviewers, which have helped us to significantly improve the quality of our manuscript. As per the professional comments and questions, the manuscript has been checked carefully and revised accordingly (using “Track Changes”). Specifically, the main modifications and point-to-point responses were listed below corresponding to the comments.
General comments:
Deng et al. developed statistical multiple regression model to understand temporal and spatial variation of concentrations of Chlorophyll-a (Chla) in Lake Taihu. They use an impressive 11-year-long data series of monthly measurements from 10 stations. They identified water temperature (WT), dissolved organic matter (DOM), total phosphorous (TP) and pH to positively correlate with Chla, and dissolved oxygen (DO), nitrate and transparency to correlate negatively. In the model they used only those variables that correlated positively. The model was developed based on data series collected at some station and validated against the other stations. The outcome of validation indicates that the model can be used for management of Lake Taihu.
The study is not novel from the ecological point of view, but I imagine that it can be useful for monitoring and management purposes, thus I think it deserves publication in IJERPH. The manuscript is well written and nice to read. The Discussion can be shorten by removing or substantially shortening section 4.1 (on factor influencing Chla concentration), as this part is not novel at all. Concerning the model, I do not understand why the variables that correlate negatively were not included in the model. If the authors calculated such a model and it turned out to be worse that the current one, it should be included in the ms, otherwise it should be explained. Finally, I also think it would be valuable to validate the model against subsequent seasonal data if they are available (years 2016-2017/2018?).
Response: Thanks for your professional suggestion. In the discussion part, some no novel explanations on effects of NPRs, DO and SD on Chl-a concentrations have been removed. The explanation why the variables correlated negatively with Chl-a concentration was not included in the model has been given in the comment 11. Regarding the data set of water quality from the years of 2016 to 2018, the relevant explanation has also been provided in the comment 7.
Minor comments:
L28: explain the abbreviation for SDResponse: Thanks for your comments. The SD is the abbreviation for Secchi depth. In the revised manuscript, we substituted Secchi depth for SD.
L111: what was the volume? This sentence is not very clear at the first reading.Response: Thanks for your comments. We rewrote “Two liters of mixed water samples from equal volumes at 10 cm, 30 cm and 50 cm below the water surface were collected and transported on ice to the laboratory for further analysis.” to “Two liters of well-mixed water samples from the 10 cm, 30 cm and 50 cm below the water surface were collected and transported on ice to the laboratory for further analysis.”
L113: DOM is defined as < 0.45 µm, thus these samples were collected incorrectly. Consider removing the DOM data from the model (see also below for comments to L291-301)Response: Thanks your comments. DOM means dissolved organic matters. Generally, DOM was determined using filtrate. In some published papers, DOM was measured using filtrate after 0.45 µm filter membranes, such as Jiang et al. (Water Research 169 (2020) 115217 ) and Jiang et al. (Bioresource Technology 292 (2019) 121942), etc. However, 1.2 μm filter membrane used to obtain filtrate containing DOM was also found in some papers, such as Derrien et al. (Science of the Total Environment 694 (2019) 133714). In our study, 0.45 µm filter membrane was used to obtain filtrate containing DOM.
L118: explain the abbreviation for TPResponse: Thank you for your comments. In our manuscript, total phosphorus first appeared on the line 72, where TP is put in a bracket.
L118-119: This sentence is a bit confusing with regard to making unclear whether the samples for DOM where filtered or not?Response: Thanks for your comment. In the revised manuscript, we rewrote “Unfiltered sample water from each sampling site was digested using potassium persulfate before measuring the concentrations of TN, TP and DOM using the method of Jin and Tu [24].” to “Unfiltered sample water from each sampling site was digested using potassium persulfate before measuring the concentrations of TN and TP, and the filtrate of raw water was used to measure DOM according to using the method of Jin and Tu [24].”
L125: how often was the sonde calibrated?Response: Thanks for your suggestion. In the revised manuscript, we rewrote “WT, pH and dissolved oxygen (DO) were monitored in situ at all sampling sites using a water quality sonde (YSI-6600V2, Yellow Spring Instruments, USA) with sensors calibrated regularly” to “WT, pH and dissolved oxygen (DO) were monitored in situ at all sampling sites using a water quality sonde (YSI-6600V2, Yellow Spring Instruments, USA) with calibrated sensors prior to use.”
L148: Would it be possible to test the model on data from subsequent years (2016+)?Response: Thanks for your suggestion. The data set of water quality from the years of 2016 to2018 was not available. Therefore, we did not test the MLR model using water quality parameters from recent 3 years.
L170-L172: I do not understand why these correlations were calculated if you did the regression. Please explain.Response: Thanks for your comment. According to the results on the significant correlations between the Chl-a concentration and other water quality, as well as principal component analysis, some key water quality parameters were screened to develop the MLR model. Consequently, the correlations between the Chl-a and water quality were calculated.
L180-L181: please write how you defined seasons. Did you mean astronomical seasons (winter: 22 Dec-21 Mar; spring 22 Mar-20 Jun; summer: 21 Jun-21 Sep; autumn: 22 Sep-21 Dec) or meteorological seasons (winter: Dec-Feb; spring Mar-May; summer: Jun-Aug; autumn: Sep-Nov).Response: Thanks for your comment. In the revised manuscript, we rewrote “Variations in the mean Chl-a levels for spring, autumn and winter were low among the 4 sampling sites but the summer values at S1 were significantly higher than those at S3 and S4.” to “Variations in the mean Chl-a levels for spring (Mar. - May), autumn (Sep. - Nov.) and winter (Dec. - Feb.) were low among the 4 sampling sites but the summer (June - Aug.) values at S1 were significantly higher than those at S3 and S4 (Figure 3).”
Figure 4: it seems that there was one outlier (Station S1, 2007), have you tested its influence on correlations, regression and the model performance?Response: Thanks for your comment. In 2007, severe cyanobacterial bloom took place in Lake Taihu, Chl-a concentration in water column was much higher than those of other years. S1 was in the north of Meiliang Bay, where cyanobacteria easily accumulated under the proper prevailing wind direction. The data set of water quality from S1 to S4 was used to develop correlation and regression analyses. The result demonstrated that the Chl-a concentration of S1 in 2007 has no significant influence on correlations, regression and the model performance.
L222-224: please explain why the variables that correlated negatively (mainly nitrogen) were not included in the model.Response: Thanks for your comment. The correlations between the Chl-a concentration and water quality were analyzed, and weak negative relationships between the Chl-a concentration and DO, NO3--N, ln(SD) were observed. Meanwhile, the principal component analysis of water quality from the S1 to S4 was carried out. The water quality variables, which significantly related to Chl-a concentrations and mainly contributed to three PCs, were screened to develop a MLR model. Therefore, the variables correlated negatively with Chl-a concentrations were not used to develop the MLR model.
L288: please explain why DO was excluded.Response: Thanks for your valuable comment. The attachment of Supplementary Materials was not be uploaded during submitting the manuscript, which caused the explanation of DO exclusion to be insufficient and confusing. In the revised manuscript, we rewrote “The eigenvalues, CI, variance proportion, tolerance and VIF, which are characteristic parameters of the collinearity diagnostics listed in Table 2 and Supporting Information, were used to determine the five water quality variables for the development of the MLR model; however, DO was excluded.” to “The eigenvalues, CI, variance proportion, tolerance and VIF, which are characteristic parameters of the collinearity diagnostics, were listed in the Supporting Information and Table 2, indicating that there was the collinearity between the constant and DO contents and thus DO was excluded from the five water quality variables for the development of the MLR model.” Additionally, the Supplementary materials were also provided.
Supplementary Materials
Collinearity statistics of the five water quality variables derived from a multiple linear regression using a stepwise method.
|
Model |
D |
Eigenvalue |
CI |
Variance Proportions |
|||||
|
Constant |
COD |
DO |
TP |
WT |
pH |
||||
|
1 |
1 |
1.948 |
1.000 |
.03 |
.03 |
|
|
|
|
|
2 |
.052 |
6.108 |
.97 |
.97 |
|
|
|
|
|
|
2 |
1 |
2.932 |
1.000 |
.00 |
.01 |
.00 |
|
|
|
|
2 |
.067 |
6.613 |
.01 |
.95 |
.01 |
|
|
|
|
|
3 |
.001 |
48.579 |
.99 |
.04 |
.99 |
|
|
|
|
|
3 |
1 |
3.739 |
1.000 |
.00 |
.00 |
.00 |
.01 |
|
|
|
2 |
.228 |
4.048 |
.00 |
.00 |
.00 |
.34 |
|
|
|
|
3 |
.031 |
10.914 |
.01 |
.86 |
.00 |
.55 |
|
|
|
|
4 |
.001 |
57.891 |
.99 |
.14 |
1.00 |
.10 |
|
|
|
|
4 |
1 |
4.585 |
1.000 |
.00 |
.00 |
.00 |
.00 |
.01 |
|
|
2 |
.241 |
4.366 |
.00 |
.00 |
.00 |
.32 |
.07 |
|
|
|
3 |
.142 |
5.685 |
.00 |
.01 |
.00 |
.01 |
.79 |
|
|
|
4 |
.031 |
12.101 |
.01 |
.84 |
.00 |
.52 |
.00 |
|
|
|
5 |
.001 |
69.079 |
.99 |
.14 |
1.00 |
.15 |
.14 |
|
|
|
5 |
1 |
5.446 |
1.000 |
.00 |
.00 |
.00 |
.00 |
.00 |
.00 |
|
2 |
.321 |
4.117 |
.00 |
.00 |
.00 |
.17 |
.01 |
.02 |
|
|
3 |
.187 |
5.395 |
.00 |
.01 |
.00 |
.08 |
.30 |
.01 |
|
|
4 |
.032 |
13.098 |
.00 |
.87 |
.00 |
.55 |
.00 |
.01 |
|
|
5 |
.013 |
20.307 |
.05 |
.01 |
.01 |
.16 |
.39 |
.78 |
|
|
6 |
.001 |
82.817 |
.95 |
.10 |
.99 |
.04 |
.30 |
.18 |
|
D denotes dimension; CI means the condition index
Figure 8: please explain in the methods how the probabilities were calculated and present these results also in the result section.Response: Thanks for your comment. In the manuscript, use of the word “probabilities” caused a unclear and confusing expression. Practically, the probabilities mean percentage of NPRs exceeding 16. In revised manuscript, percentage was used and the result of NPRs was also added in the discussion part.
L291-301: This paragraph should more discuss that phytoplankton also produces and releases DOM, even if it’s not major DOM source in the Lake Taihu. This maybe the reason for the observed strong correlation (r2>0.5). Moreover, as mentioned before, samples for DOM measurements were not correctly collected in my opinion. This need to be critically address or DOM should be remove from the model.Response: Thanks for your valuable comments. In the discussion part, we add some description on DOM source in eutrophic freshwater lakes, although qualitative contribution of various DOM sources to algal growth was not distinguished according water quality. Concerning the filter method of DOM has been explained in the comment 3.
L324: The lake Taihu seems to be very shallow and large, so I imagine that it easily mixes even during summer. How likely are the occurrences of thermal stratification and how long does it last?Response: Thanks for your professional comment. In the revised manuscript, the description on the effect of thermal stratification in shallow lakes on the Chl-a concentrations was removed.
L354-355: “The trophic status at S5 to S8 is somewhat similar to that of S1 to S4” Could you be more specific here.Response: Thanks for your comment. We rewrote “The trophic status at S5 to S8 is somewhat similar to that of S1 to S4” to “The trophic statuses at S5 to S8 belong to eutrophic levels, similar to those of S1 to S4.”

Round 2
Reviewer 1 Report
I believe that the the revised manuscript is meeting the criteria to be published. However, I have a minor comment to ask, regarding the additional Supplementary Materials you provided:
The COD abbreviation/parameter is used in the table, please explain why?
Generally, I believe that this manuscript is a nice contribution to the filed of limnological modelling and I recommend to be published, after answering this minor comment.
Author Response
We are grateful for your comment, carefulness and patience, which help us to improve the manuscript. Regarding the COD used in the additional Supplementary Materials was incorrectly spelled, the correct parameter should be DOM. We are quite regretful for providing the confusing information. We will correct it and re-submit the the additional Supplementary Materials.
Supporting material
Collinearity statistics of the five water quality variables derived from a multiple linear regression using a stepwise method.
|
Model |
D |
Eigenvalue |
CI |
Variance Proportions |
|||||
|
Constant |
DOM |
DO |
TP |
WT |
pH |
||||
|
1 |
1 |
1.948 |
1.000 |
.03 |
.03 |
|
|
|
|
|
2 |
.052 |
6.108 |
.97 |
.97 |
|
|
|
|
|
|
2 |
1 |
2.932 |
1.000 |
.00 |
.01 |
.00 |
|
|
|
|
2 |
.067 |
6.613 |
.01 |
.95 |
.01 |
|
|
|
|
|
3 |
.001 |
48.579 |
.99 |
.04 |
.99 |
|
|
|
|
|
3 |
1 |
3.739 |
1.000 |
.00 |
.00 |
.00 |
.01 |
|
|
|
2 |
.228 |
4.048 |
.00 |
.00 |
.00 |
.34 |
|
|
|
|
3 |
.031 |
10.914 |
.01 |
.86 |
.00 |
.55 |
|
|
|
|
4 |
.001 |
57.891 |
.99 |
.14 |
1.00 |
.10 |
|
|
|
|
4 |
1 |
4.585 |
1.000 |
.00 |
.00 |
.00 |
.00 |
.01 |
|
|
2 |
.241 |
4.366 |
.00 |
.00 |
.00 |
.32 |
.07 |
|
|
|
3 |
.142 |
5.685 |
.00 |
.01 |
.00 |
.01 |
.79 |
|
|
|
4 |
.031 |
12.101 |
.01 |
.84 |
.00 |
.52 |
.00 |
|
|
|
5 |
.001 |
69.079 |
.99 |
.14 |
1.00 |
.15 |
.14 |
|
|
|
5 |
1 |
5.446 |
1.000 |
.00 |
.00 |
.00 |
.00 |
.00 |
.00 |
|
2 |
.321 |
4.117 |
.00 |
.00 |
.00 |
.17 |
.01 |
.02 |
|
|
3 |
.187 |
5.395 |
.00 |
.01 |
.00 |
.08 |
.30 |
.01 |
|
|
4 |
.032 |
13.098 |
.00 |
.87 |
.00 |
.55 |
.00 |
.01 |
|
|
5 |
.013 |
20.307 |
.05 |
.01 |
.01 |
.16 |
.39 |
.78 |
|
|
6 |
.001 |
82.817 |
.95 |
.10 |
.99 |
.04 |
.30 |
.18 |
|
D denotes dimension; CI means the condition index